# The consequences of using different epoch lengths on the classification of accelerometer based sedentary behaviour and physical activity

Teatske M. Altenburg[1]*, Xinhui Wang[1,2,3], Evi van Ekris[1], Lars Bo Andersen[4], Niels Christian Møller[5], Niels Wedderkopp[5,6], Mai J. M. Chinapaw[1]

1 Department of Public and Occupational Health, Amsterdam UMC, Vrije Universiteit Amsterdam, Amsterdam Public Health research institute, Amsterdam, The Netherlands, 2 Luxembourg Centre for Systems Biomedicine, University of Luxembourg, Belvaux, Luxembourg, 3 College of Computer Science, Qinghai Normal University, Xining, Qinghai, China, 4 Faculty of Education, Arts and Sports, Campus Sogndal, Western Norway University of Applied Sciences, Bergen, Norway, 5 Department of Sports Science and Clinical Biomechanics, Centre of Research in Childhood Health, University of Southern Denmark, Odense, Denmark, 6 Orthopedic Department, Institute of Regional Health Services Research, University of Southern Denmark, Hospital of Middelfart, Odense, Denmark

☯ These authors contributed equally to this work.
* t.altenburg@amsterdamumc.nl

**Data Availability Statement:** For data from the SOS-study: All relevant data are within the manuscript and its Supporting information files.

## Abstract

We examined the influence of using different epoch lengths on the classification accuracy of laboratory-controlled sedentary behaviour (SB), and free-living total time and time spent in bouts of SB and moderate-to-vigorous physical activity (MVPA), in children and adolescents. We used two studies including accelerometer-derived data of: 1) controlled activities, i.e. seven sedentary, one standing and one dancing (n = 90); 2) free-living activities (n = 902). For the controlled-activity data, we calculated percentages of time classified as SB and MVPA. For the free-living data, we calculated medians (25th–75th percentiles) of total time and time spent in bouts of SB and MVPA. Applying 8counts/5seconds, 25counts/15seconds and 100counts/60seconds for SB on controlled-activity data revealed respectively (1) 92–96%, 89–99% and 98–100% of sedentary time accurately classified as SB (activity- and age-dependent); (2) 91–98%, 88–99% and 97–100% of standing time classified as SB (age-dependent); (3) 25–37%, 20–25% and 25–38% of dancing time classified as SB (age-dependent). Using longer epochs, children's total time in SB and MVPA decreased while time accumulated in bouts of SB and MVPA accumulated in bouts increased. We conclude that a 60-second epoch seems preferable when the aim is to classify sedentary behaviour, while a shorter epoch length is needed to capture children's short bursts of MPVA. Furthermore, we should be aware that a longer epoch results in averaging of intensities to the middle category.

For data from the CHAMPS-study: The data supporting the conclusions of this article are stored in the Danish National Archives (https://www.sa.dk/en/research-researchers-research-service-the-danish-national-archives/use-the-danish-national-archives-survey-data/). Data are available upon request from the CHAMPS-DK steering committee, as the data as determined by the University's legal department is not allowed to be stored on a public server due to legal and ethical restrictions. Interested parties may contact the Danish National Archive (mailboxDDA@sa.dk) and Dr. Niels Wedderkopp (nwedderkopp@health.sdu.dk). The following information is mandatory at the time of application: a description of how data will be used, securely managed and stored, and finally permanently deleted.

**Funding:** The contributions of Altenburg, Wang, van Ekris and Chinapaw were funded by the Netherlands Organization for Health Research and Development (Grant number 91211057). The contribution of Wang was further funded by the National Natural Science Foundation of China (NSF 61263039 and 11101321) and the Qinghai Science & Technology Department Project (QHSTDP 2017-ZJ-768 and QHSTDP 2018-ZJ-776). We would like to thank prof Niels Wedderkopp University of Southern Denmark for providing us data from the CHAMPS study. The CHAMPS-study was funded by: The TRYG Foundation, University College Lillebaelt, University of Southern Denmark, The Nordea Foundation, The IMK foundation, The Region of Southern Denmark, The Egmont Foundation, The A.J. Andersen Foundation, The Danish Rheumatism Association, Østifternes Foundation, Brd. Hartmann's Foundation, TEAM Denmark, The Danish Chiropractor Foundation, The Nordic Institute of Chiropractic and Clinical Biomechanics. The funding organisations had no role in this study. The funders had no role in study design, data collection and analysis, decision to publish, or preparation of the manuscript.

**Competing interests:** The authors have declared that no competing interests exist.

# Introduction

Accelerometers are widely used to assess time spent in physical activity and sedentary behaviour in children and adolescents. In recent years, accelerometers have been advanced to collect second-by-second activity count data, instead of collecting data over a pre-defined interval known as epoch length. Nevertheless, when applying cut-points to classify accelerometer data into sedentary behaviour and physical activity of different intensities, data is aggregated in specified epoch lengths. Studies validating cut-points for sedentary behaviour, light, moderate and vigorous physical activity (LPA, MPA and VPA, respectively) generally applied epoch lengths ranging from 15 to 60 seconds [1–7]. Despite cut-points being established in a specific epoch length, they are adjusted to match to other epoch lengths. For example, the Puyau cut-point of ≥3200 counts per minute for classifying moderate-to-vigorous physical activity (MVPA) can be adjusted to match a 15-second epoch, resulting in a cut-point of ≥800 counts per 15-seconds. A reason for applying shorter epoch lengths is children's sporadic activity pattern, which may be best captured in shorter epoch lengths [8,9]. While the epoch length for collecting data is generally well-reported, most studies in children do not report the epoch length for analysing data. As cut-points are generally referred to as 'counts per minute' it could be assumed that a 60-second epoch is applied [10].

A number of studies examined the influence of applying different epoch lengths on estimates of children's and adolescents' time spent in sedentary behaviour, LPA, MPA and MVPA [9,11–19]. Overall, these studies demonstrated significantly less sedentary time [13,14,16,19–22], more LPA [13,14,16,19–22] and less VPA [11,13–16,19–22] with increasing epoch lengths. For time spent in MPA and MVPA lower, higher and similar time estimates were found with increasing epoch lengths [11–17,19–22].

To date, few studies examined the influence of epoch length on time estimates of physical activity [11,16,17,19] or sedentary behaviour [16,19] accumulated in bouts in children and adolescents. With increasing epoch lengths, Aibar et al [17] found less time in MVPA bouts of ≥10 minutes, whereas Nilsson et al [11] found more time in MPA bouts of ≥10 minutes. Nettlefold et al [16] and Aadland et al [19] examined the influence of epoch length on bouts of physical activity and sedentary behaviour of different durations. Both studies found more time accumulated in MVPA bouts of various durations with increasing epoch lengths, i.e. more time in MVPA bouts ranging between 5 and ≥20 minutes [16] and VPA and MVPA bouts ranging between 3 and 29 minutes [19]. Additionally, both studies found more time accumulated in bouts of sedentary behaviour and LPA with increasing epoch lengths, especially in bouts ranging from 5–10 minutes [16] and 5–29 minutes [19]. The differences in the influence of epoch length on time accumulated in bouts of MPA, VPA and MVPA might be attributed to differences in definitions of these bouts. Previous studies differed in the duration of bouts of sedentary behaviour and physical activity and the allowance of tolerance time within these bouts. Nilsson et al [11] did not allow tolerance within MPA and VPA bouts, Aibar et al [17] allowed 1 or 2 minutes below the MPA threshold, and Nettlefold et al [16] allowed 30 seconds tolerance within MVPA bouts of 5–10 minutes (15-second epoch analysis only), 1 minute within bouts of 10–20 minutes and 2 minutes within bouts of ≥20 minutes. Aadland et al [19] examined bouts of sedentary behaviour and physical activity without and with tolerance time, with tolerance time up to 20% of the bout duration, equalling a maximum interruption of 4 minutes in a 20-minute bout. Importantly, laboratory studies in young adults have demonstrated that such brief interruptions in sedentary time may already affect cardiometabolic health indicators [23–27]. Hence, a bout of sedentary time is previously defined as a period of uninterrupted sedentary behaviour [28–30]. For MVPA, an allowance of up to 10% tolerance

time within a bout is justified, as during such short interruptions the metabolic rate may still be above the threshold for MVPA.

McClain et al [12] examined the combined effects of applying different epoch lengths and cut-points on accelerometer-derived MVPA, compared to direct observation, in children monitored during physical education (PE). Deviations in accelerometer-derived estimates were larger with higher epoch lengths when using cut-points of Treuth (i.e. 3000 counts per minute) and Mattocks (i.e. 3581 counts per minute) for MVPA classification but not when using Freedson's cut-point (i.e. 1910 and 2060 counts per minute for 10- and 11-year-olds, respectively), compared to direct observation. The 5-second epoch resulted in the smallest deviations from observation-based MVPA, for all cut-points [12]. This finding illustrates that it is more difficult to accumulate MVPA using longer epochs as this activity intensity should be continued for a longer period, i.e. short bursts of MVPA may not be detected. By including direct observation as comparison measure, McClain et al [12] could not only examine differences in accelerometer-derived MVPA, but also conclude on the most accurate epoch length for assessing MVPA. To date, studies examining the influence of epoch length on the classification accuracy of accelerometer-derived sedentary behaviour are lacking.

The aims of the present study were to examine the influence of using different epoch lengths (i.e. 5, 15 and 60 seconds) on (1) the classification accuracy of controlled sedentary activities in children and adolescents and (2) estimates of children's free-living sedentary behaviour and physical activity, both for total time as well as time accumulated in bouts.

## Materials and methods

### Design and participants

For this study we used two exising datasets. For the first aim (i.e. epoch length influence on classification accuracy of controlled sedentary activities), data from the SOS (sick of sitting) study was used (for details see van Ekris et al [31]). In short, the SOS-study is a controlled laboratory study examining body movement during a wide range of sedentary activities, standing and dancing among children and adolescents. Fifty-three children aged 10–12 years and 37 adolescents aged 16–18 years from three primary and two secondary schools in and around Amsterdam participated in the SOS-study. All participants followed a standardized protocol including one dancing, one standing and seven sedentary activities, i.e. lying down, drawing/writing, playing sedentary computer games, sitting without any imposed activity, using a tablet, watching movies and media multitasking (i.e. playing on a tablet while watching movies). Dancing was included as a control activity to check whether body movements during dancing were different from body movements during sedentary activities and standing. Participants were requested to avoid standing still during dancing, implying that dancing was performed at a light intensity at the lowest. Participants completed the standardized protocol at school or at the VU University Medical Center. Each activity lasted 10 minutes and was followed by a short break (i.e. 3–5 minutes). All activities were supervised by trained researchers and video recorded using four cameras (AIPTEK AHD H125, AIPTEK International Inc., Taiwan), to confirm compliance with the protocol. Epochs during which children did not comply to the protocol, e.g. non-sedentary activity during sedentary activities or sedentary behaviour during the standing and dancing activity, were removed from the data. The study protocol of the SOS-study (number 13/031) was approved by the VU University Medical Center Ethical Committee. Written informed consent was signed by one parent (participants aged <18) or the participant (participants aged 18).

For the second aim (i.e. epoch length influence on free-living sedentary behaviour and physical activity), baseline data from the CHAMPS-study DK was used (for details see [32]). In

short, the CHAMPS study is a natural experiment evaluating the effect of additional PE lessons in children attending public primary schools in the municipality of Svendborg in Denmark. From the six schools that were willing to participate, all children and parents from kindergarten to 4th grade (age range 5.5–12 years) were invited to participate. In total, 697 (90%) children and parents from intervention schools and n = 521 (71%) from control schools agreed to participate. For the present study, we used baseline accelerometer data (Nov 2009/Jan 2010). A total of 902 children had valid accelerometer data and were included in the analysis. The CHAMPS-study was carried out in accordance with the Declaration of Helsinki, and registered in the Danish Data Protection Agency (J.nr. 2008-41-2240). Parents of the children gave written informed consent and the children gave verbal consent and could at any time withdraw from the study.

## Measurements

Physical activity and sedentary time were assessed using GT3X ActiGraph accelerometers (Pensacola, Florida, USA) using the vertical axis and standard filtering using a 1-second epoch in the SOS-study and a 2-second epoch in the CHAMPS-study. The accelerometer was placed at the right hip of the participants by the research staff using customized elastic belts. Participants in the SOS-study wore the accelerometer during the standardized protocol. Participants in the CHAMPS-study were instructed to wear the device from the time they woke up in the morning until bedtime for seven consecutive days, except for when engaging in water-based activities. Additionally, SOS-participants wore a Polar RS800CX heart rate monitor (Polar Electro Oy, Kempele, Finland) that was attached to an adjustable strap and tied around their chest, to obtain an indication of the intensity of the dancing activity. The intensity of the dancing activity was calculated by expressing children's heart rate (HR, in beats per minute (bpm)) during dancing as percentage of their heart rate reserve (HRR), using the formula: dancing HR = [% HRR x (maximum HR–resting HR)] + resting HR [33]. Maximum HR was predicted by the formula: 208-(0.7 x age) [34]. Resting HR was defined as the lowest HR measured during lying. Moderate intensity was defined as at least 40% HRR [35].

## Accelerometer data reduction

Second-by-second accelerometer data (SOS-study) and accelerometer data collected in 2-second epochs (CHAMPS-data) were analysed using a customized software program developed in R based on the data reduction recommendations by Chinapaw et al [36]. First, data were transferred into 5-, 15- and 60-second epoch lengths. CHAMPS-data collected in 2-second epochs were transferred into 5- and 15-second epochs by summing the counts from respectively the first two epochs (i.e. first 4 seconds) and the first 7 epochs (i.e. first 14 seconds), and half of the counts of the subsequent 2-second epoch (i.e. 5th and 15th second, respectively). For the SOS-study, the first three minutes of each 10-minute activity were excluded from the data analysis ensuring steady state of the activity (i.e. leaving 7-minute data in the analysis for each activity). For inclusion in the data analysis of the CHAMPS-study, each participant needed a minimum of six and a maximum of 7 different days, with at least eight valid hours per day. Non-wear time was defined as periods of more than 60 minutes of consecutive zero counts and excluded from analysis [36]. Sedentary time was defined as each 5-, 15- and 60-second epoch below 8, 25 and 100 counts, respectively [5,6,37,38]. The Evenson cut-points were used to define LPA, MPA, VPA and MVPA [5]. Table 1 presents an overview of the Evenson cut-points for sedentary time and physical activity, adjusted to match 5-, 15- and 60-second epochs. We defined a sedentary bout as a period of at least 10 minutes of uninterrupted sedentary time [29,30] and distinguished between bouts of 10–29.9 and ≥30 minutes. We defined a

**Table 1. Evenson cut-points for sedentary behaviour and physical activity adjusted to match epochs of 5, 15 and 60 seconds.**

|  | 5-second epoch (i.e. counts 5 s$^{-1}$) | 15-second epoch (i.e. counts 15 s$^{-1}$) | 60-second epoch (i.e. counts min$^{-1}$) |
|---|---|---|---|
| SB | 0–8 | 0–25 | 0–100 |
| LPA | 8–191 | 26–573 | 101–2295 |
| MPA | 191–334 | 574–1002 | 2296–4011 |
| VPA | ≥334 | ≥1003 | ≥4012 |
| MVPA | ≥192 | ≥574 | ≥2296 |

Abbreviations: LPA, light physical activity; min, minute; MPA, moderate physical activity; MVPA, moderate-to-vigorous physical activity; SB, sedentary behaviour; s, seconds; VPA, vigorous physical activity.

physical activity bout as a period of at least 10 minutes of LPA or at least 5 minutes of MPA, VPA and MVPA, allowing 10% below the lower threshold with an absolute tolerance of 3 consecutive minutes to prevent too much time below the specified cut-point [39]. Moreover, for physical activity bouts, we allowed an unlimited amount of time above the upper threshold, when this time was not already captured in a PA bout of higher intensity [39]. For example, when VPA time was not captured in a VPA bout of 5–9.9 or ≥10 minutes, it could be included as tolerance time in an MPA or LPA bout. We distinguished bouts of 10–29.9 and ≥30 minutes for LPA, and bouts of 5–9.9, 10–19.9, 20–29.9 and ≥30 minutes for MPA, VPA and MVPA. Periods less than 10 minutes for sedentary behaviour and LPA and periods less than 5 minutes for MPA, VPA and MVPA were referred to as 'sporadic'. The software program extracted VPA bouts first, followed by MPA, LPA and sedentary bouts; MVPA bouts were extracted in a separate analysis. As a consequence of tolerance allowance within VPA, MPA, MVPA and LPA bouts, total time accumulated in bouts does not count up to the total time spent in VPA and MPA.

## Statistical analyses

Statistical analysis were performed in SPSS (version 18.0) and R. Descriptive participant characteristics were presented as means (standard deviation), for both samples. Using SOS-data, we determined the accuracy of classifying sedentary behaviour by calculating the percentage of time accelerometer counts were below the sedentary cut-point, using 5-, 15- and 60-second epochs, respectively. Since we did not measure energy expenditure, we could not calculate the accuracy of classifying MVPA for the dancing activity. Therefore, we present the percentage of time accelerometer counts were above the MVPA cut-point, using in 5-, 15- and 60-second epochs, respectively. As we previously demonstrated that body movement during sedentary activities, standing and dancing were different for children and adolescents [31], we conducted all SOS-data analyses for children and adolescents separately. Using CHAMPS-data, as time estimates according to 5-, 15- and 60-second epoch lengths were not normally distributed, medians (25th–75th percentiles) were calculated. To analyse the influence of epoch length, data were log-transformed (log2) and subsequently tested for significance in R using one-way repeated measures ANOVA (anova_test, rstatix package, R platform 3.4.0; $p < 0.05$).

## Results

Children (43% girls) and adolescents (49% girls) from the SOS-study were on average 12.0 (SD = 0.8) and 17.4 (SD = 0.4) years old, respectively. Children (56% girls) in the CHAMPS-study were on average 9.4 (SD = 1.4) years old.

**Table 2. Percentage of time classified as sedentary behaviour and MVPA for the different activities, using different epoch lengths.**

| | Lying down | Drawing | Gaming | Watching movies | Tablet use | Media multi-tasking | Sitting | Standing | Dancing |
|---|---|---|---|---|---|---|---|---|---|
| **Children (n = 53)** | | | | | | | | | |
| % of time classified as SED | | | | | | | | | |
| $\leq$8cnts 5sec$^{-1}$ | 96.0 | 92.3 | 96.3 | 95.3 | 95.2 | 95.6 | 92.3 | 90.6 | 36.7 |
| $\leq$25cnts 15sec$^{-1}$ | 93.6 | 91.2 | 96.0 | 92.8 | 94.0 | 92.1 | 88.5 | 88.4 | 24.8 |
| $\leq$100cnts min$^{-1}$ | 98.1 | 98.6 | 100 | 97.8 | 99.7 | 99.7 | 99.2 | 97.0 | 38.2 |
| % of time classified as MVPA | | | | | | | | | |
| $\geq$191cnts 5sec$^{-1}$ | 0.3 | 0.2 | 0 | 0.2 | 0 | 0.1 | 0 | 0.2 | 20.1 |
| $\geq$574cnts 15sec$^{-1}$ | 0.2 | 0 | 0 | 0.1 | 0 | 0.1 | 0 | 0.4 | 20.1 |
| $\geq$2296cnts min$^{-1}$ | 0 | 0 | 0 | 0 | 0 | 0 | 0 | 0 | 0 |
| **Adolescents (n = 37)** | | | | | | | | | |
| % of time classified as SED | | | | | | | | | |
| $\leq$8cnts 5sec$^{-1}$ | 99.2 | 99.4 | 99.3 | 98.8 | 99.1 | 98.5 | 97.8 | 98.1 | 25.2 |
| $\leq$25cnts 15sec$^{-1}$ | 98.9 | 99.5 | 99.1 | 98.0 | 98.4 | 98.5 | 96.6 | 98.7 | 19.9 |
| $\leq$100cnts min$^{-1}$ | 100 | 100 | 100 | 100 | 99.2 | 100 | 100 | 100 | 25.1 |
| % of time classified as MVPA | | | | | | | | | |
| $\geq$191cnts 5sec$^{-1}$ | 0 | 0 | 0 | 0 | 0 | 0 | 0.1 | 0 | 29.5 |
| $\geq$574cnts 15sec$^{-1}$ | 0 | 0 | 0 | 0 | 0 | 0 | 0 | 0 | 33.0 |
| $\geq$2296cnts min$^{-1}$ | 0 | 0 | 0 | 0 | 0 | 0 | 0 | 0 | 0 |

Abbreviations: Cnts, counts; min, minute; MVPA, moderate-to-vigorous physical activity; sec, seconds.

## Influence of different epoch lengths on the classification accuracy of children's and adolescents' controlled activities (SOS-study)

Table 2 shows the percentage of children's and adolescents' time classified as sedentary, LPA and MVPA for the different sedentary activities, standing and dancing. Applying 8 counts per 5 seconds for sedentary time and $\geq$191 counts per 5 seconds for MVPA resulted in accurate classification of sedentary time between 92.3% (sitting and drawing) and 96.3% (gaming) of the time for children and between 97.8% (sitting) and 99.4% (drawing) of the time for adolescents. All other 5-second epochs in adolescents and almost all other 5-second epochs in children were classified as LPA. In children, a small percentage of time in lying (0.3%), drawing (0.2%), watching movies (0.2%) and media multi-tasking (0.1%) was classified as MVPA. Standing was classified as sedentary time in 90.6% of time in children and 98.1% of time in adolescents. Dancing was classified as sedentary time for 36.7% of children's time and 25.2% of adolescents' time, and mainly classified as LPA (children: 43.2% LPA and 20.1% MVPA; adolescents: 45.3% LPA and 29.5% MVPA). During dancing, heart rate was on average 33% ($\pm$ 14) for children's and 40% ($\pm$ 18) for adolescents' heart rate reserve. According to heart rate, 34.5% of children's dancing time and 48.5% of adolescents' dancing time was classified as of moderate intensity.

Applying 25 counts per 15 seconds for sedentary time and $\geq$574 counts per 15 seconds for MVPA resulted in accurate classification of sedentary time between 88.5% (sitting) and 96.0% (gaming) of the time for children and between 96.6 (sitting) and 99.5% (drawing) for adolescents. All other 15-second epochs were classified as LPA, except for a small percentage of time in lying (0.2%), watching movies (0.1%) and media multi-tasking (0.1%) in children. For standing, 88.4% (children) and 98.7% (adolescents) of the time was classified as sedentary, and 0.4% (children) and 0% (adolescents) of the time was classified as MVPA. For dancing, 24.8% of children's and 19.9% of adolescents' dancing time was classified as sedentary time. Dancing

time was mainly classified as LPA in children and adolescents (children: 55.1% LPA and 20.1% MVPA; adolescents: 47.1% LPA and 33.0% MVPA).

Applying 100 counts per minute for sedentary time and ≥2296 counts per minute for MVPA resulted in accurate classification of sedentary time between 98.0% (watching movies) and 100% (gaming) of the time for children. Adolescents' sedentary activities were accurately classified 100% of time, except for tablet use (99.2%). All other 60-second epochs during the sedentary activities were classified as LPA. Standing was classified as sedentary for 97.0% of time in children and 100% of time in adolescents. Dancing was classified as sedentary for 38.2% of children's and 25.1% of adolescents' time. None of the dancing time was classified as MVPA, indicating that dancing was mainly classified as LPA (i.e. 61.8% in children and 74.9% in adolescents).

## Influence of epoch length on estimates of children's free-living activities (CHAMPS-study)

Table 3 and Fig 1 show the median (25th and 75th percentiles) duration and frequency of total time and time in bouts of physical activity and sedentary behaviour using 5-, 15-and 60-second epochs. Children's median wear time was similar across the different epochs, i.e. 817, 817 and 818 minutes per day using a 5-, 15- and 60-second epoch, respectively. Using longer epochs, total (533, 471 and 385 minutes per day for 5-, 15- and 60-sec epochs, respectively) and sporadic (455, 358 and 194 minutes per day, respectively) sedentary time decreased, while sedentary time in bouts of 10–30 minutes (51, 78 and 127 minutes per day, respectively) and ≥30 minutes (0, 0 and 35 minutes per day, respectively) increased. Total LPA and LPA in bouts was higher with increasing epoch lengths (total time: 196, 259 and 353 minutes per day for 5-, 15- and 60-second epochs, respectively; 10-30-minute bouts: 24, 55 and 110 minutes per day, respectively; ≥30-minute-bouts: 0, 32 and 108 minutes per day, respectively) whereas sporadic LPA was highest using a 15-second epoch (201 minutes per day) and lowest during a 60-second epoch (161 minutes per day). Total and sporadic MVPA decreased using longer epochs (total time: 60, 55 and 44 minutes per day for 5-, 15- and 60-second epochs, respectively; sporadic time: 42, 23 and 6 minutes per day, respectively) whereas time in MVPA bouts slightly increased (visible from 90th percentiles: 10-30-minute bouts: 8.8, 13.9 and 17.4 minutes per day, respectively; ≥30-minute-bouts: 29.2, 29.8 and 136.5 minutes per day, respectively). Also, the number of bouts of sedentary behaviour and physical activity increased with increasing epoch lengths whereas sporadic accumulations of sedentary behaviour and physical activity decreased.

## Discussion

This study examined the influence of different epoch lengths (i.e. 5, 15 and 60 seconds) on (1) the classification accuracy of controlled sedentary activities in children and adolescents; and (2) estimates of free-living sedentary time and time spent in physical activity at different intensities in children, both for total time and time spent in bouts of different durations.

To the best of our knowledge, this is the first study examining the accuracy of classifying sedentary time applying different epoch lengths in children and adolescents. The classification accuracy was especially low using a 5- and 15-second epoch to classify the activities 'drawing' and 'sitting only' in children, of which 8% of time was classified as LPA. For the 'sitting only' activity this underestimation of sedentary time using a short epoch length can be explained by the observation from video recordings that children were 'moving' a lot on their chair during this activity (i.e. sitting restlessly), which may be evoked by a lack of a concrete task. The 60-second epoch has masked the relative high accelerometer counts evoked by this restlessly

**Table 3. Prevalence (duration and frequency; median [25th-75th percentiles]) of total and bouts sedentary behaviour and physical activity using different epoch lengths (n = 902 children).**

| TOTAL DAY | Min/day | | | Nr/day | | |
|---|---|---|---|---|---|---|
| | 5-sec | 15-sec | 60-sec | 5-sec | 15-sec | 60-sec |
| Wear time | 816.8 [710.8; 881.5] | 817.0 [711.0; 881.8] | 818.0 [712.0; 882.0]^ | N/A | N/A | N/A |
| **SB** | | | | | | |
| 0–9.9 min[1] | 455.3 [376.8; 518.5] | 357.8 [292.5; 414.5] | 194.0 [154.0; 233.0]^ | 717.0 [586.0; 842.0] | 289.0 [237.0; 341.0] | 68.0 [54.0; 82.0]^ |
| 10–29.9 min | 51.2 [24.1; 90.8] | 78.0 [43.3; 124.5] | 127.0 [84.0; 177.0]^ | 4.0 [2.0; 6.0] | 5.0 [3.0; 8.0] | 8.0 [6.0; 11.0]^ |
| ≥30 min | 0.0 [0.0; 38.9] | 0.0 [0.0; 45.0] | 35.0 [0.0; 146.0]^ | 0.0 [0.0; 1.0] | 0.0 [0.0; 1.0] | 1.0 [0.0; 2.0]^ |
| total min | 533.4 [449.0; 616.0] | 471.0 [390.0; 556.8] | 385.0 [308.0; 473.0]^ | N/A | N/A | N/A |
| **LPA** | | | | | | |
| 0–9.9 min[1] | 177.7 [143.0; 210.3] | 200.8 [163.0; 238.3] | 161.0 [127.0; 192.0]^ | 899.0 [733.5; 1059.0] | 334.0 [273.0; 390.0] | 112.0 [93.0; 126.0]^ |
| 10–29.9 min[2] | 23.8 [0.0; 46.3] | 54.8 [28.5; 85.0] | 110.0 [78.0; 146.0]^ | 1.0 [0.0; 3.0] | 3.0 [2.0; 5.0] | 7.0 [5.0; 9.0]^ |
| ≥30 min[2] | 0.0 [0.0; 0.0] | 32.3 [0.0; 60.0] | 108.0 [52.0; 176.0]^ | 0.0 [0.0; 0.0] | 1.0 [0.0; 1.0] | 2.0 [1.0; 4.0]^ |
| total min | 196.1 [158.8; 232.7] | 259.5 [210.3; 306.3] | 353.0 [287.0; 413.0]^ | N/A | N/A | N/A |
| **MPA** | | | | | | |
| 0–4.9 min[1] | 26.8 [19.5; 34.8] | 17.5 [12.3; 23.5] | 6.0 [3.0; 10.0]^ | 220.0 [161.0; 285.52] | 50.0 [36.0; 68.0] | 5.0 [3.0; 7.0]^ |
| 5–9.9 min[2] | 0.0 [0.0; 0.0] | 0.0 [0.0; 0.0] | 0.0 [0.0; 0.0]^ | 0.0 [0.0; 0.0] | 0.0 [0.0; 0.0] | 0.0 [0.0; 0.0]^ |
| ≥10 min[2] | 0.0 [0.0; 0.0] | 0.0 [0.0; 0.0] | 0.0 [0.0; 11.0]^ | 0.0 [0.0; 0.0] | 0.0 [0.0; 0.0] | 0.0 [0.0; 1.0]^ |
| total min | 35.5 [25.3; 47.8] | 37.0 [24.3; 51.8] | 32.0 [18.0; 52.0] ^ | N/A | N/A | N/A |
| **VPA** | | | | | | |
| 0–4.9 min[1] | 14.7 [9.0; 21.9] | 5.3 [2.8; 9.0] | 1.0 [0.0; 3.0]^ | 105.0 [68.0; 150.0] | 15.0 [9.0; 23.0] | 1.0 [00; 2.0]^ |
| 5–9.9 min[2] | 0.0 [0.0; 0.0] | 0.0 [0.0; 0.0] | 0.0 [0.0; 0.0]^ | 0.0 [0.0; 0.0] | 0.0 [0.0; 0.0] | 0.0 [0.0; 0.0]^ |
| ≥10 min[2] | 0.0 [0.0; 0.0] | 0.0 [0.0; 0.0] | 0.0 [0.0; 0.0]^ | 0.0 [0.0; 0.0] | 0.0 [0.0; 0.0] | 0.0 [0.0; 0.0]^ |
| total min | 23.3 [13.3; 37.0] | 16.3 [7.8; 29.3] | 8.0 [2.0; 19.0]^ | N/A | N/A | N/A |
| **MVPA** | | | | | | |
| 0–4.9 min[1] | 42.2 [29.6; 56.2] | 23.0 [16.0; 31.8] | 6.0 [3.0; 11.0]^ | 242.0 [177.0; 314.0] | 54.0 [39.0; 72.0] | 5.0 [3.0; 8.0]^ |
| 5–9.9 min[2] | 0.0 [0.0; 0.0] | 0.0 [0.0; 0.0] | 0.0 [0.0; 0.0]^ | 0.0 [0.0; 0.0] | 0.0 [0.0; 0.0] | 0.0 [0.0; 0.0]^ |
| ≥10 min[2] | 0.0 [0.0; 0.0] | 0.0 [0.0; 0.0] | 0.0 [0.0; 18.0]^ | 0.0 [0.0; 0.0] | 0.0 [0.0; 0.0] | 0.0 [0.0; 1.0]^ |
| total min | 59.9 [40.4; 84.6] | 54.8 [34.3; 80.8] | 44.0 [23.0; 71.0]^ | N/A | N/A | N/A |

Abbreviations: LPA, light physical activity; min, minutes; MPA, moderate physical activity; MVPA, moderate-to-vigorous physical activity; N/A, not applicable; SB, sedentary behaviour; sec, seconds.

[1]Indicates sporadic activity.

[2]For LPA bouts of at least 10 minutes and MPA, VPA and MVPA bouts of at least 5 minutes, we allowed tolerance time within a bout: 10% tolerance was allowed above the upper threshold and 10% tolerance below the lower threshold, with the latter restricted to an absolute tolerance of 3 consecutive minutes. Therefore, time spent in these bouts do not count up to the total time in the various behaviours.

^Indicates a significant influence of epoch length.

sitting, as there is more room (i.e. time) for compensation. When using a 5- and 15-second epoch, slight movements during sedentary activities are more easiliy picked up as LPA, while a 60-second epoch is less sensitive to such interruptions.

Strikingly, for all epoch lengths we found that a large percentage of dancing time was classified as sedentary, in both age groups but especially in children. This may partly be explained by children and adolescents occasionally standing still for small periods of time during the dancing activity, to choose and start a new dance. When using a short epoch length, these short standing periods during the dancing activity were picked up as sedentary behaviour as accelerometers do not distinguish standing still from sitting still. As we did not mark these short standing periods and these periods might not necessarily have led to a lower heart rate (i.e. based on heart rate: 34% and 49% of dancing time classified as moderate intensity), we

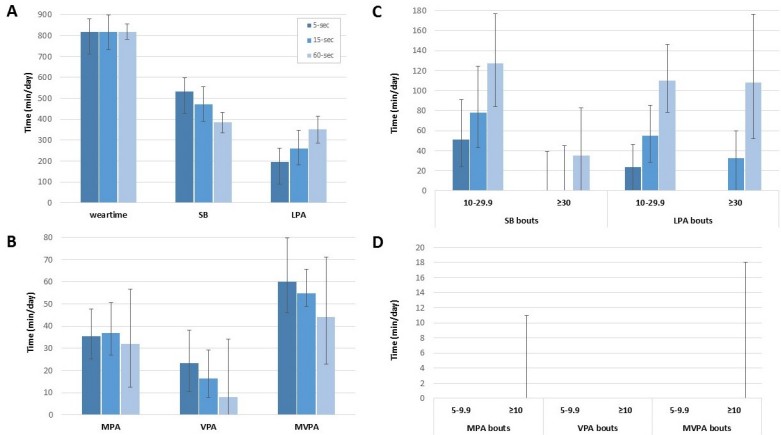

**Fig 1. Estimates of total time and time accumulated in bouts of sedentary behaviour and physical activity.** Total time estimates of A: Total wear time, sedentary behaviour (SB) and light physical activity (LPA), and B: Total moderate, vigorous and moderate-to-vigorous physical activity (MPA, VPA and MVPA); and total time estimates of time spent in C: Bouts of 10–29.9 and ≥30 minutes of SB and LPA; and D: Bouts of 5–9.9 and ≥10 minutes of MPA, VPA and MVPA, applying cut-points that are adjusted to match epoch lengths of 5, 15 and 60 seconds.

could not assess whether the accelerometer accurately classified these periods of not being at a moderate intensity. However, zero percent of dancing time was classified as MVPA when applying a cut-point of ≥2296 counts per minute, while applying a cut-point of ≥574 counts per 15 seconds resulted in 20% (children) and 33% (adolescents) of dancing time classified as MVPA, and applying ≥192 counts per 5 seconds resulted in 20% (children) and 29 (adolescents) of dancing time classified as MVPA. This indicates that for children and adolescents a 60-second epoch lacks the sensitivity to detect short bursts of MVPA during dancing. In contrast to our finding, McClain et al [12] showed that the classification accuracy of MVPA was not influenced by epoch length (range: 10- to 60-second epochs) when applying the Freedson cut-point (4 MET; ≥1910 and 2060 counts per minute for 10- and 11-year-olds, respectively), which is similar to the Evenson cut-point we applied (≥2296 counts per minute). An explanation for these different findings may be the physical activities included in the different study protocols. The SOS-study included only one physical activity (i.e. dancing) which based on heart rate was classified as moderate intensity 34–49% of time, and which was occasionally interrupted with standing still. Apparently this dancing activity included short periods of moderate intensity activity, which were detected using 5- and 15-second epochs but not using a 60-second epoch. The children in the study of McClain et al [12] participated in a range of physical activities during PE class. The activities during PE may have induced a more constant level of moderate-to-vigorous intensity of longer duration detected by the applied 10-60-second epoch lengths.

Our results from the SOS-study showed that applying cut-points of ≤8 counts per 5 seconds and ≤25 counts per 15 seconds underestimated sedentary time, however, applying these cut-points to free-living data (i.e. data from CHAMPS-study) resulted in larger estimates of total sedentary time than a cut-point of ≤100 counts per minute. Nevertheless, this finding from the SOS-study (i.e. smaller estimates of sedentary time with increasing epoch lenght) is in line with previous studies [13,16,20–22]. Surprisingly, *total* time spent in LPA was 44% and 26% lower, when applying a 5- and 15-second epoch, respectively, than a 60-second epoch. The inconsistent findings between controlled and free-living sedentary time, and the large differences between LPA estimates across epochs lengths may suggest that a substantial amount of free-living *total* LPA was classified as sedentary behaviour using 5- and 15-second epochs.

Previous laboratory-based studies applied epoch lengths of 15-seconds [5,7], 30-seconds [1] or 60-seconds [37] to establish and evaluate the 100 counts per minute sedentary cut-point. One limitation of these studies is that they only focused on the accuracy of classifying sedentary activities, without examining whether light activities would be misclassified as sedentary. We recommend future studies evaluating the classification accuracy of cut-points and the potential influence of epoch length to include activities covering the full spectrum of sedentary behaviour and physical activity at various intensities.

We found less sporadic and total sedentary time with increasing epoch length, which is partly in line with findings of Aadland et al [19] and Nettlefold et al [16]. Our finding of increasing sedentary time in bouts of 10–30 and ≥30 minutes with longer epochs is in line with findings of Nettlefold et al [16]. Applying longer epochs resulted in less LPA time in bouts of 10–30 and ≥30 minutes. In contrast, Aadland et al [19] found that LPA time was especially accumulated in longer bouts (i.e. bouts ≥5 minutes compared to bouts <5 minutes) when applying a 60-second vs. a 1- and 10 second epoch.

Based on data from the CHAMPS-study, accelerometer-derived estimates of free-living VPA and MVPA were lower whereas estimates of MPA were similar with increasing epoch lengths. Previous studies found lower, higher and similar estimates of total MPA and MVPA time [11–17,19,21,22] and lower estimates of total VPA time [11,13–16,19,21,22] with increasing epoch lengths. Differences between studies may be related to differences in data reduction procedures, e.g. definition of non-wear time and cut-points used to define activity levels. Banda et al [21] demonstrated that wear time estimates varied by epoch length when applying a wear time algorithm using an allowance period. However, the influence of epoch length on time estimates of sedentary behaviour, LPA, MPA, VPA and MVPA was similar across wear time algorithms. Although not explicitly examined, data from the supplementary tables provided by Banda et al. [21] suggest that the influence of epoch length varies by different cut-points. Unfortunately, previous studies did not always report details on data reduction procedures, limiting the comparison between studies. We recommend future studies to clearly report all data reduction procedures, including the epoch length in the specific data reduction steps, to increase comparison between studies.

With increasing epoch length, we found a slightly longer duration and slighty higher frequency of bouts of MPA, VPA and MVPA, both for 5-10-minute and ≥10-minute bouts (visible in the 90th percentiles). It should be noted that these differences were small due to the low occurrence of these bouts of MPA, VPA and MVPA. Four previous studies examined the influence of epoch length on bouts of MPA and MVPA [11,16,17,19]. Aibar et al [17] and Aadland et al [19] found lower estimates of MVPA in bouts of ≥10 minutes [17] and lower estimates of MPA, VPA and MVPA in bout durations between 1.5–1.9 minutes and ≥60 minutes [19] when applying higher epoch lengths (i.e. 60 versus 3 seconds [17] or 60 versus 1 seconds [19]). Nettlefold et al [16] found higher estimates of MVPA in bouts of 5–10, 10–20 and ≥20 minutes and Nilsson et al [11] found higher MPA estimates in bouts of ≥10 minutes when applying higher epoch lengths. Differences in findings between studies may be explained by differences in wear time definitions (i.e. ≥10, 30 or 60 minutes consecutive zeros, with and without allowance) and cut-points used to classify physical activity levels (i.e. Evenson [5], Freedson [40] and Trost [41]). Moreover, differences may be explained by different tolerance times in bouts, as demonstrated by Aadland et al [19]. In the present study we allowed tolerance above (unlimited) and below (10%, up to an absolute maximum of 3 consecutive minutes) the thresholds for LPA, MPA and VPA. As a result, for example, a bout of MPA may contain up to 10% LPA. The consequence of allowing tolerance is that the time spent in bouts does not count up to the total time in physical activity. Logically, the amount of tolerance time within physical activity bouts is higher when using a larger epoch length and with longer bout

durations. Thus, although allowing tolerance time within a physical activity bout is reasonable from a physiological perspective, it complicates interpretability of the time estimates of physical activity bouts. Currently, there is no consensus on the definition of physical activity bouts, yet this is urgently needed to allow comparison between studies.

One strength of our study includes the combination of examining estimates of sedentary behaviour and physical activity during controlled activities as well as during free-living activities. Examining the effects of epoch length on total time as well as time accumulated in bouts and both at various intensities ranging from sedentary behaviour to VPA further strengthens our study. A limitation is the inclusion of only one dancing activity in our laboratory study, performed at a light to moderate intensity, according to heart rate. Secondly, as accelerometers cannot distinguish between standing and sitting, we cannot comment on the accuracy of classifying acceleromer-based free-living sedentary time and LPA. Third, the use of heart rate monitoring to estimate the intensity of the dancing activity is a limitation of our study, as heart rate provides a delayed record of the physiological response.

## Conclusions

We conclude that a 60-second epoch seems preferable when the aim is to classify accelerometer-based sedentary behaviour, while shorter epochs are needed to capture children's sporadic moderate or vigorous physical activity intensity. Furthermore, we should be aware that a longer epoch results in averaging of intensities to the middle category.

## Supporting information

**S1 File. Data SOS-study based on 5-, 15- and 60-second epochs.**
(XLSX)

## Author Contributions

**Conceptualization:** Teatske M. Altenburg, Lars Bo Andersen, Niels Christian Møller, Niels Wedderkopp, Mai J. M. Chinapaw.

**Data curation:** Evi van Ekris.

**Formal analysis:** Teatske M. Altenburg, Xinhui Wang, Evi van Ekris.

**Funding acquisition:** Teatske M. Altenburg, Lars Bo Andersen, Niels Christian Møller, Niels Wedderkopp, Mai J. M. Chinapaw.

**Methodology:** Teatske M. Altenburg, Xinhui Wang, Lars Bo Andersen, Niels Christian Møller, Niels Wedderkopp, Mai J. M. Chinapaw.

**Project administration:** Teatske M. Altenburg.

**Software:** Xinhui Wang.

**Supervision:** Teatske M. Altenburg.

**Visualization:** Teatske M. Altenburg.

**Writing – original draft:** Teatske M. Altenburg.

**Writing – review & editing:** Xinhui Wang, Evi van Ekris, Lars Bo Andersen, Niels Christian Møller, Niels Wedderkopp, Mai J. M. Chinapaw.

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
