## [Decision Letter · Decision Letter 0]

6 Apr 2021

PONE-D-20-40338

The consequences of using different epoch lengths on the classification of accelerometer based sedentary behaviour and physical activity

PLOS ONE

Dear Dr. Altenburg,

Thank you for submitting your manuscript to PLOS ONE. After careful consideration, we feel that it has merit but does not fully meet PLOS ONE’s publication criteria as it currently stands. Therefore, we invite you to submit a revised version of the manuscript that addresses the points raised during the review process.

Your paper was reviewed by two subject experts, who also reviewed the previously submitted version of your paper. As you will see below, both still have some major concerns about the paper which must be addressed ahead of resubmission. Of particular importance, Reviewer 2 suggests your conclusions do not support the data presented in the paper. Please ensure this observation is fully considered in your response. 

We look forward to receiving your revised manuscript.

Kind regards,

Kathryn L. Weston, PhD

Academic Editor

PLOS ONE

Journal Requirements:

Reviewers' comments:

Reviewer's Responses to Questions

**Comments to the Author**

1. Is the manuscript technically sound, and do the data support the conclusions?

Reviewer #1: Partly

Reviewer #2: No

2. Has the statistical analysis been performed appropriately and rigorously? 

Reviewer #1: No

Reviewer #2: Yes

3. Have the authors made all data underlying the findings in their manuscript fully available?

Reviewer #1: No

Reviewer #2: Yes

4. Is the manuscript presented in an intelligible fashion and written in standard English?

Reviewer #1: Yes

Reviewer #2: Yes

5. Review Comments to the Author

Reviewer #1: I thank the authors for providing good answers to my comments and making appropriate revisions to the manuscript. Though the manuscript has improved considerably, I have some more comments, partly as a follow-up of my previous comments, that need to be considered.

1) Regarding the focus on epoch length or cut point scaling, this is obviously two sides of the same coin. I thank the authors for revising their title to reflect the focus on epoch durations, which I believe is a much easier conceptualization than rescaling of cut points, since (as argued previously) cut points obviously must be scaled when varying the epoch setting. However, besides the title, the rest of the manuscript largely focus on rescaling of cut points. Please consider improving the readability further by focusing on the epoch setting throughout. The rescaling of the cut points is a consequence of the analysis of the different epoch settings and can be described as is in the methods.

2) One more issue of conceptualization. The manuscript focuses throughout on the misclassification of epochs and intensities of PA. If it can be verified by the criterion (video observation) that every single epoch is in fact of the prescribed intensity this makes sense. However, given the application of three different types of information regarding PA/intensity classification, that is, observation, heart rate, and accelerometry, which do not capture the same information, I ask the authors to consider whether “misclassification” could be conceptualized differently. Instead of stating that short epochs misclassified dancing time as sedentary, I would say that a short epoch setting has increased sensitivity for detecting sedentary bouts during dancing. Since short periods of standing still during dancing were not marked, I would argue this finding is not a misclassification. From a perspective of classifying movement (not energy consumption), these periods are probably correctly classified. For sedentary activities, this issue is more complex. Though the accelerometers probably correctly have identified movement (acceleration), as stated line 346-, this time could possibly still be classified as sedentary given the definition of being seated. Please note that if longer periods of activity by default is classified at a specific intensity (as done herein), longer epochs will obviously be best suited to classify because variations/interruptions are ignored by the criterion. Please consider to reconceptualize to moderate the conclusions, or at least elaborate on this issue (as partly already done) in the discussion.

3) It is concluded that a 15-second epoch setting should be used for analysis of PA. This conclusion contrasts the findings by Aadland et al (your ref 19), showing that every second counts, and the findings by Aadland et al (https://doi.org/10.1080/02640414.2019.1693320), showing that a 1-sec epoch setting provided better information of the relationship between PA and cardiometabolic health than 10- and 60-sec epoch settings, thus leading the authors to conclude a 1-sec epoch should be applied. Please include the latter study to inform the discussion and conclusion. Related to the comment above, these findings probably result from increased sensitivity to capture short burst of activity throughout the day.

4) Please consider and correct the description of how 5-sec epochs were constructed line 183-. I guess this text should state “the first 2 epochs (i.e. first 4 seconds) and the first 7 epochs (i.e. first 14 seconds), and half of the counts of the subsequent 2-second epoch (i.e. 5th and 15th second, respectively). ”?

5) The Man Whitney U test is a test for two independent samples. Given that the data on the different three epoch settings derive from the same individuals, a repeated measures test suitable for three “timepoints” should be applied.

6) The results seem focused on comparison of two epoch settings, e.g., line 293- (“using longer epochs, total (533 vs 385 min/day) …”), but must reflect the three settings now included. Please correct.

7) There are few typos, e.g., line 459 (hear). Please correct.

Reviewer #2: Dear authors

Thank you for responding to our request to include also 5-second epochs. By doing that, it is more apparent that your study does not support your conclusions, and actually is in favor for a shorter epoch, i.e. 5 seconds.

Firstly, in the controlled study, you have a priory assigned 8 activities as sedentary and dancing as physical activity across the 7 minutes included. If the individual moves spontaneously during this (as they may not be able to be still; I have seen this many times in my own studies), this will be captured more easily by the 5-second epoch but at the same time it is evaluated as miss-classification due to your a priory categorization. In fact, the 5-second epoch tells what the child actually is doing. By using a 60-second epoch, the spontaneous variation is “filtered” away. Adolescents probably have more ability to be still and your results show that the misclassification in this age-group was very low. Further, based on the lowest amount of time as sedentary during dancing for 15-second epochs you draw the conclusion that this is the preferable epoch. However, as you also pointed out in the second paragraph on page 20, dancing may not be a continuous MVPA activity, but rather more variated.

Secondly, your free-living study demonstrate the same pattern as in for example Aadland et al, that with increasing epoch length there is less total time SB, more total time LPA, slightly less total time MPA and less total time VPA. This pattern is the result of, that with less allowance for variation, the closer the value come to the middle intensities LPA and MPA. In addition, LPA and MPA are very broad categories, which will easily “take over” the time with increasing epoch length. Further, when you restrict data by bouts, it is even more difficult for the variation that is captured by the 5-second to be inside the boundary of the intensity category. Hence, that is why you see the reduction of SB with increasing epoch length using the shortest bout (0-9.9 min), which is more similar to the total time, and for increasing bout length as well.

My conclusion would be the opposite to yours. I find it a little puzzling in your controlled study that the 5-second epoch is more similar to the 60-second epoch for % of time classified as SED, than to the 15-second epoch.

6. PLOS authors have the option to publish the peer review history of their article (what does this mean?). If published, this will include your full peer review and any attached files.

Reviewer #1: **Yes: **Eivind Aadland

Reviewer #2: No

---

## [Author Response · Author response to Decision Letter 0]

27 May 2021

REVIEWER#1 

I thank the authors for providing good answers to my comments and making appropriate revisions to the manuscript. Though the manuscript has improved considerably, I have some more comments, partly as a follow-up of my previous comments, that need to be considered.

AUTHORS' REPLY

We thank the reviewer for acknowledging that our revised the manuscript answers his previously raised questions, and appreciate the critical review of our revised manuscript. Please find our reply to the follow-up questions below.

1) Regarding the focus on epoch length or cut point scaling, this is obviously two sides of the same coin. I thank the authors for revising their title to reflect the focus on epoch durations, which I believe is a much easier conceptualization than rescaling of cut points, since (as argued previously) cut points obviously must be scaled when varying the epoch setting. However, besides the title, the rest of the manuscript largely focus on rescaling of cut points. Please consider improving the readability further by focusing on the epoch setting throughout. The rescaling of the cut points is a consequence of the analysis of the different epoch settings and can be described as is in the methods.

AUTHOR’S REPLY

We have revised the manuscript now focusing on applying different epoch lengths (vs. scaling cut-points) as the reviewer suggested. 

2) One more issue of conceptualization. The manuscript focuses throughout on the misclassification of epochs and intensities of PA. If it can be verified by the criterion (video observation) that every single epoch is in fact of the prescribed intensity this makes sense. However, given the application of three different types of information regarding PA/intensity classification, that is, observation, heart rate, and accelerometry, which do not capture the same information, I ask the authors to consider whether “misclassification” could be conceptualized differently. Instead of stating that short epochs misclassified dancing time as sedentary, I would say that a short epoch setting has increased sensitivity for detecting sedentary bouts during dancing. Since short periods of standing still during dancing were not marked, I would argue this finding is not a misclassification. From a perspective of classifying movement (not energy consumption), these periods are probably correctly classified. For sedentary activities, this issue is more complex. Though the accelerometers probably correctly have identified movement (acceleration), as stated line 346-, this time could possibly still be classified as sedentary given the definition of being seated. Please note that if longer periods of activity by default is classified at a specific intensity (as done herein), longer epochs will obviously be best suited to classify because variations/interruptions are ignored by the criterion. Please consider to reconceptualize to moderate the conclusions, or at least elaborate on this issue (as partly already done) in the discussion.

AUTHOR’S REPLY

We indeed applied video observation as our criterion measure, and we checked for all epochs whether the participants complied to the protocol, i.e. we deleted non-sedentary activity during sedentary activities and sedentary activity during standing and dancing activities (see description of the SOS-study in the methods section; page 6-7; lines 136-141). 

Participants wore a heart rate monitor only to be able to obtain an indication of the intensity of the dancing activity. We have clarified this in the methods section (page 7; lines 167-169): “Additionally, SOS-participants wore a Polar RS800CX heart rate monitor (Polar Electro Oy, Kempele, Finland) that was attached to an adjustable strap and tied around their chest, to obtain an indication of the intensity of the dancing activity.” 

Based on the above, we argue that the short periods of standing can be referred to as misclassifications of sedentary behaviour as our video observations revealed that participants were not in a seated posture. Our finding reflects the limitation that accelerometers are not able to differentiate between standing still and sitting still. However, we acknowledge the viewpoint of the reviewer and therefore 1) changed the wording “misclassified” into the neutral "classified” in the results section; and 2) elaborated on this issue in our discussion section:

- We have deleted the paragraph in the discussion where we relied on these ‘misclassifications’ to inform our conclusion (pages 19-20; lines 331-340 of previous version). 

- We revised/added the following sentences to our discussion section:

 o Page 20; lines 341-343: “When using a 5- and 15-second epoch, slight movements during sedentary activities are more easily picked up as LPA, while a 60 second epoch is less sensitive for such interruptions.”

 o Page 20; lines 348-350: “When using a short epoch length, these short standing periods during the dancing activity were picked up as sedentary behaviour as accelerometers do not distinguish standing still from sitting still.”

 o Page 20; lines 354-360: “However, zero percent of dancing time was classified as MVPA when applying a cut-point of ≥2296 counts per minute, while applying a cut-point of ≥574 counts per 15 seconds resulted in 20% (children) and 33% (adolescents) of dancing time classified as MVPA, and applying ≥192 counts per 5 seconds resulted in 20% (children) and 29 (adolescents) of dancing time classified as MVPA. This indicates that for children and adolescents a 60-second epoch lacks the sensitivity to detect short bursts of MVPA during dancing.” 

3) It is concluded that a 15-second epoch setting should be used for analysis of PA. This conclusion contrasts the findings by Aadland et al (your ref 19), showing that every second counts, and the findings by Aadland et al (https://doi.org/10.1080/02640414.2019.1693320), showing that a 1-sec epoch setting provided better information of the relationship between PA and cardiometabolic health than 10- and 60-sec epoch settings, thus leading the authors to conclude a 1-sec epoch should be applied. Please include the latter study to inform the discussion and conclusion. Related to the comment above, these findings probably result from increased sensitivity to capture short burst of activity throughout the day.

AUTHOR’S REPLY

We agree with the reviewer and have revised our discussion and conclusion sections accordingly.

- We have deleted the paragraph in the discussion where we relied on these misclassifications to inform our conclusion (pages 19-20; lines 331-340 of previous version).

- We revised/added the following sentences to our discussion section:

 o Page 20; lines 341-343: “When using a 5- and 15-second epoch, slight movements during sedentary activities are more easily picked up as LPA, while a 60 second epoch is less sensitive for such interruptions.”

 o Page 20; lines 348-350: “When using a short epoch length, these short standing periods during the dancing activity were picked up as sedentary behaviour as accelerometers do not distinguish standing still from sitting still.”

 o Page 20; lines 354-360: “However, zero percent of dancing time was classified as MVPA when applying a cut-point of ≥2296 counts per minute, while applying a cut-point of ≥574 counts per 15 seconds resulted in 20% (children) and 33% (adolescents) of dancing time classified as MVPA, and applying ≥192 counts per 5 seconds resulted in 20% (children) and 29 (adolescents) of dancing time classified as MVPA. This indicates that for children and adolescents a 60-second epoch lacks the sensitivity to detect short bursts of MVPA during dancing.” 

- We have rephrased our conclusion, in the abstract and main text:

 o Page 2; lines 40-42: “We conclude that a 60-second epoch seems preferable when the aim is to classify sedentary behaviour, while a shorter epoch length is needed to capture children’s short bursts of MPVA.”

 o Page 24; lines 455-457: “We conclude that a 60-second epoch seems preferable when the aim is to classify accelerometer-based sedentary behaviour, while shorter epochs are needed to capture children’s sporadic moderate or vigorous intensity physical activity.”

4) Please consider and correct the description of how 5-sec epochs were constructed line 183-. I guess this text should state “the first 2 epochs (i.e. first 4 seconds) and the first 7 epochs (i.e. first 14 seconds), and half of the counts of the subsequent 2-second epoch (i.e. 5th and 15th second, respectively)”?

AUTHOR’S REPLY

We thank the reviewer for pointing at this error. We have corrected this sentence as suggested.

5) The Man Whitney U test is a test for two independent samples. Given that the data on the different three epoch settings derive from the same individuals, a repeated measures test suitable for three “timepoints” should be applied.

AUTHOR’S REPLY

We have adjusted this and conducted one-way repeated measures ANOVA, using log-transformed data:

- P11; lines 237-240: “To analyse the influence of epoch length, data were log-transformed and subsequently tested for significance in R using one-way repeated measures ANOVA (anova_test, rstatix package, R platform 3.4.0; p<0.05).“

- P18; line 317: “^Indicates a significant influence of epoch length.”

Based on this analysis all variables showed a significant influence of epoch length, which we indicated in Table 3. As the description of the results was already based on relevance rather than significance, there was no need to revise the description of our results (i.e. page 16).

6) The results seem focused on comparison of two epoch settings, e.g., line 293- (“using longer epochs, total (533 vs 385 min/day) …”), but must reflect the three settings now included. Please correct.

AUTHOR’S REPLY

We have revised this paragraph, now including the results for the three epoch settings (results, page 16). For example (page 16, lines 292-296): 

“Using longer epochs, total (533, 471 and 385 minutes per day for 5-, 15- and 60-sec epochs, respectively) and sporadic (455, 358 and 194 minutes per day, respectively) sedentary time decreased, while sedentary time in bouts of 10-30 minutes (51, 78 and 127 minutes per day, respectively) and ≥30 minutes (0, 0 and 35 minutes per day, respectively) increased.” 

7) There are few typos, e.g., line 459 (hear). Please correct.

AUTHOR’S REPLY

Thank you for pointing us at this typo. We have carefully checked our manuscript for typos.

REVIEWER#2: Dear authors

Thank you for responding to our request to include also 5-second epochs. By doing that, it is more apparent that your study does not support your conclusions, and actually is in favor for a shorter epoch, i.e. 5 seconds.

AUTHORS' REPLY

We thank the reviewer for his/her valuable comments, which we addressed below.

Firstly, in the controlled study, you have a priory assigned 8 activities as sedentary and dancing as physical activity across the 7 minutes included. If the individual moves spontaneously during this (as they may not be able to be still; I have seen this many times in my own studies), this will be captured more easily by the 5-second epoch but at the same time it is evaluated as miss-classification due to your a priory categorization. In fact, the 5-second epoch tells what the child actually is doing. By using a 60-second epoch, the spontaneous variation is “filtered” away. Adolescents probably have more ability to be still and your results show that the misclassification in this age-group was very low. Further, based on the lowest amount of time as sedentary during dancing for 15-second epochs you draw the conclusion that this is the preferable epoch. However, as you also pointed out in the second paragraph on page 20, dancing may not be a continuous MVPA activity, but rather more variated.

AUTHOR’S REPLY

We agree with the reviewer that the accelerometer might have captured all movements of the children, which is filtered away when using a 60-second epoch length. We revised our manuscript including the conclusion as follows:

- We have deleted the paragraph in the discussion where we relied on these misclassifications to inform our conclusion (pages 19-20; lines 331-340 of previous version).

- We revised/added the following sentences to our discussion section:

 o Page 20; lines 341-343: “When using a 5- and 15-second epoch, slight movements during sedentary activities are more easily picked up as LPA, while a 60 second epoch is less sensitive for such interruptions.” Additionally, we have changed the wording “misclassified” into the neutral “classified” (results section).

 o Page 20; lines 348-350: “When using a short epoch length, these short standing periods during the dancing activity were picked up as sedentary behaviour as accelerometers do not distinguish standing still from sitting still.”

 o Page 20; lines 354-360: “However, zero percent of dancing time was classified as MVPA when applying a cut-point of ≥2296 counts per minute, while applying a cut-point of ≥574 counts per 15 seconds resulted in 20% (children) and 33% (adolescents) of dancing time classified as MVPA, and applying ≥192 counts per 5 seconds resulted in 20% (children) and 29 (adolescents) of dancing time classified as MVPA. This indicates that for children and adolescents a 60-second epoch lacks the sensitivity to detect short bursts of MVPA during dancing.” 

- We have rephrased our conclusion, in the abstract and main text:

 o Page 2; lines 40-42: “We conclude that a 60-second epoch seems preferable when the aim is to classify sedentary behaviour, while a shorter epoch length is needed to capture children’s short bursts of MPVA.”

 o Page 24; lines 455-457: “We conclude that a 60-second epoch seems preferable when the aim is to classify accelerometer-based sedentary behaviour, while shorter epochs are needed to capture children’s sporadic moderate or vigorous intensity physical activity.”

Secondly, your free-living study demonstrate the same pattern as in for example Aadland et al, that with increasing epoch length there is less total time SB, more total time LPA, slightly less total time MPA and less total time VPA. This pattern is the result of, that with less allowance for variation, the closer the value come to the middle intensities LPA and MPA. In addition, LPA and MPA are very broad categories, which will easily “take over” the time with increasing epoch length. Further, when you restrict data by bouts, it is even more difficult for the variation that is captured by the 5-second to be inside the boundary of the intensity category. Hence, that is why you see the reduction of SB with increasing epoch length using the shortest bout (0-9.9 min), which is more similar to the total time, and for increasing bout length as well.

AUTHOR’S REPLY

We thank the reviewer for this clear observation and added this to our conclusion:

- Page 24; lines 457-458: “Furthermore, we should be aware that a longer epoch results in averaging of intensities to the middle category.”

My conclusion would be the opposite to yours. I find it a little puzzling in your controlled study that the 5-second epoch is more similar to the 60-second epoch for % of time classified as SED, than to the 15-second epoch.

AUTHOR’S REPLY

We agree with the reviewer that the similarity between the 5- and 60-second epoch for the % of time classified as sedentary behaviour is puzzling. We have no explanation for this.

We also agree with the reviewer that our conclusion regarding the recommendation of a 15-second epoch length when analysing physical activity was not correct, and have revised our conclusion accordingly:

- Page 2; lines 40-43: “We conclude that a 60-second epoch seems preferable when the aim is to classify sedentary behaviour, while a shorter epoch length is needed to capture children’s short bursts of MPVA. Furthermore, we should be aware that a longer epoch results in averaging of intensities to the middle category.”

- Page 24; lines 454-456: “We conclude that a 60-second epoch seems preferable when the aim is to classify accelerometer-based sedentary behaviour, while shorter epochs are needed to capture children’s sporadic moderate or vigorous intensity physical activity.”

---

## [Decision Letter · Decision Letter 1]

2 Jul 2021

The consequences of using different epoch lengths on the classification of accelerometer based sedentary behaviour and physical activity

PONE-D-20-40338R1

Dear Dr. Altenburg,

We’re pleased to inform you that your manuscript has been judged scientifically suitable for publication and will be formally accepted for publication once it meets all outstanding technical requirements.

Kind regards,

Kathryn L. Weston, PhD

Academic Editor

PLOS ONE

Additional Editor Comments (optional):

Reviewers' comments:

Reviewer's Responses to Questions

**Comments to the Author**

1. If the authors have adequately addressed your comments raised in a previous round of review and you feel that this manuscript is now acceptable for publication, you may indicate that here to bypass the “Comments to the Author” section, enter your conflict of interest statement in the “Confidential to Editor” section, and submit your "Accept" recommendation.

Reviewer #1: All comments have been addressed

Reviewer #2: All comments have been addressed

2. Is the manuscript technically sound, and do the data support the conclusions?

Reviewer #1: (No Response)

Reviewer #2: Partly

3. Has the statistical analysis been performed appropriately and rigorously? 

Reviewer #1: (No Response)

Reviewer #2: Yes

4. Have the authors made all data underlying the findings in their manuscript fully available?

Reviewer #1: (No Response)

Reviewer #2: Yes

5. Is the manuscript presented in an intelligible fashion and written in standard English?

Reviewer #1: (No Response)

Reviewer #2: Yes

6. Review Comments to the Author

Reviewer #1: I thank the authors for proving acceptable answers to my comments and making appropriate revisions to their manuscript. I have no further comments.

Reviewer #2: Dear authors

Although I find your responses to my comments reasonable, I still do not agree with your conclusions that 60s epochs are preferable for SED and shorter epochs for MPA or VPA. If the activity is pre-decided and the data processing is set to match that precondition, of course the classification accuracy increases, even if the true activity is not captured. However, this is probably more of a limitation in the traditional accelerometer methodology and with newer methodology we need to calibrate new accelerometer settings in order to classify activities more correct. Therefore, it is very important to be critical to the measurement errors caused by old methodology and not contribute to their maintenance, but instead promote improvements.

7. PLOS authors have the option to publish the peer review history of their article (what does this mean?). If published, this will include your full peer review and any attached files.

Reviewer #1: **Yes: **Eivind Aadland

Reviewer #2: No

---

## [Editor Report · Acceptance letter]

6 Jul 2021

PONE-D-20-40338R1 

The consequences of using different epoch lengths on the classification of accelerometer based sedentary behaviour and physical activity 

Dear Dr. Altenburg:

I'm pleased to inform you that your manuscript has been deemed suitable for publication in PLOS ONE. Congratulations! Your manuscript is now with our production department. 

Kind regards, 

on behalf of

Dr. Kathryn L. Weston 

Academic Editor

PLOS ONE